# OpenReview forum: "Improved Techniques for Optimization-Based Jailbreaking on Large Language Models"
_ICLR.cc/2025/Conference — ICLR 2025 Poster_

### Official Review · Reviewer_r229 · 2024-10-29

**Soundness:** 2
**Presentation:** 3
**Contribution:** 2
**Rating:** 5
**Confidence:** 3

**Summary:**

This work proposes I-GCG, an optimization-based jailbreak to improve ASR and efficiency. Compared to current methods, I-GCG formulates its optimization goal by integrating harmful information into the standard template. Also, they introduce an automatic multi-coordinate updating strategy that selects the top-p single-token suffix candidates to generate multi-token suffix candidates. Finally, the authors incorporate an easy-to-hard initialization mechanism, initially targeting a simpler jailbreaking task and then transferring the optimized suffix to more challenging tasks. Experimental results demonstrate the superior ASR and reduced time cost compared to other jailbreaks against several LLMs.

**Strengths:**

1. The authors present specific examples (Figs. 1-2 and the highlighted sentences on page 5) to emphasize the limitations of current jailbreaks that rely on a single target template, thereby distinguishing this work from previous studies.
2. This paper is well-structured, with a clear motivation and a methodology that is easy to follow.
3. The study includes recent jailbreaks published in 2023 and 2024 for comparison. The results show a high ASR and reduced time costs, supporting the contributions claimed in this paper.

**Weaknesses:**

1. This paper presents an incremental improvement over existing works. While the three proposed techniques differ from previous jailbreaks, their novelty is limited.
2. The reasons behind the improvements in jailbreak effectiveness and efficiency from the reformulated optimization goal and multi-coordinate updating strategy are not thoroughly analyzed.
3. Fig. 5 illustrates different levels of difficulty associated with malicious questions for successful jailbreak attempts. However, this result is presented solely for LLAMA2-7B-CHAT, which weakens the motivation for using the easy-to-hard initialization approach, as the difficulty of jailbreaking questions may vary on other models. Besides, this approach incurs extra time costs by requiring the jailbreaking of an easy task, raising the question of whether allocating extra time to a simple task can actually result in greater time savings on a complex task.
4. I-GCG achieves high ASR on only one dataset (AdvBench). Given the straightforward nature of the methodology, experiments on additional datasets and models would provide a more comprehensive assessment.

**Questions:**

1. In equation 6 and 7, $x^S(0)$ and $x_0^S$ serve as initial jailbreak suffixes for the start of optimization. It is unclear why the authors formulate them as constraints.
2. The analysis of the difficulty of jailbreaking different questions, particularly whether this difficulty remains consistent across other models, is lacking.
3. It is unclear whether allocating extra time to a simple task can actually result in greater time savings on a complex task.
4. Jailbreaking results on additional models, such as LLAMA2-7B alongside LLAMA2-7B-CHAT, are worth discussing. Experimental results on more datasets could further validate the effectiveness of the proposed method.
5. Providing further elaboration on the contributions of this work would be beneficial.

**Details Of Ethics Concerns:**

The proposed jailbreak may pose risks to real-world LLM systems. I recommend including a discussion on the ethical considerations.

---

> ### Author Response · Authors · 2024-11-20
> **Rebuttal by Authors [1/2]**
>
> Thank you for your valuable review and suggestions. Below we respond to the comments in **Weaknesses (W)** and **Questions (Q)**.
>
> ---
>
> ***W1 & Q5: The novelty is limited. Providing further elaboration on the contributions of this work would be beneficial.***
>
> We respectfully disagree with the assessment that our novelty is limited. As one of the first works to improve optimization-based jailbreaks for large language models (LLMs), we have systematically revealed the limitations of existing approaches like Greedy Coordinate Gradient (GCG) and proposed novel techniques to address them. Specifically:
>
> - **Expanded Scope of Target Templates:** While existing GCG methods rely on a single target template (e.g., "Sure"), we identified that this restricts attack diversity and effectiveness. Our approach introduces diverse target templates incorporating harmful self-suggestions and guidance, broadening the scope of possible attacks and showcasing the adaptability of LLM jailbreaks.
>
> - **Optimization Innovation:** From an optimization perspective, we identified inefficiencies in GCG’s single-coordinate update mechanism. By introducing an automatic multi-coordinate updating strategy—where the number of token replacements is adaptively determined—we significantly accelerate convergence without sacrificing attack performance. This innovation is both practical and generalizable, contributing to the understanding of efficient adversarial optimization for LLMs.
>
> - **Improved Initialization and Convergence Techniques:** Our proposed easy-to-hard initialization leverages a self-transfer technique that guides the attack process progressively, making it more robust and efficient. This methodological advancement is not only novel but also paves the way for more sophisticated jailbreak approaches in the future.
>
> - **Practical Impact and Extensive Evaluation:** We have conducted a thorough evaluation of our proposed $\\mathcal{I}$-GCG method on established benchmarks, including the NeurIPS 2023 Red Teaming Track. The results demonstrate that our method achieves a near 100% attack success rate, surpassing existing state-of-the-art attacks and establishing a new benchmark in the field.
>
> Our work not only improves the efficiency of optimization-based jailbreak techniques but also provides practical tools for evaluating LLM vulnerabilities. We hope that reviewers can reassess our contributions from the perspective of advancing LLM safety research and robustness testing.
>
> ---
>
> ***W2: The reasons behind the improvements in jailbreak effectiveness and efficiency from the reformulated optimization goal and multi-coordinate updating strategy.***
>
> As for the efficiency of **the reformulated optimization goal**, we have adopted the benign guidance, namely ``my response is safe to use`` and ``my output is secured``, to study the impact of harmful guidance. The results are shown in $\\textrm{\\color{blue}Table 8}$ (Page 10). Using ``My output is secured`` drops the ASR of $\\mathcal{I}$-GCG from 100% to 88%, performing worse than the I-GCG without any guidance. The results show the effectiveness of the proposed harmful guidance.
>
> As for the efficiency of **the multi-coordinate updating strategy**, we provide the following analysis: GCG uses random coordinate descent (CD) [1], which means that a single coordinate is randomly selected and updated in each iteration; in contrast, our $\\mathcal{I}$-GCG uses random block coordinate descent (BCD) [2*], where multiple coordinates are updated during each iteration. Theoretical bounds for CD and BCD are mostly applied to convex functions. Let $f(x)$ be a smooth and convex function and we aim to minimize $f(x)$, the convergence rate of CD is: $$ f(x\^{(k)})-f(x\^{\\dagger}) \\leq \\frac{2nL}{k+2nL} ||x\^{(0)}-x\^{\\dagger}||\^{2}\\textrm{,} $$ where $n$ is the number of coordinates, $L$ is a Lipschitz constant, $x\^{(k)}$ is the iterate at step $k$, and $x\^{\\dagger}$ is the optimal solution for minimizing $f(x)$. For BCD methods, the convergence rates depend on similar assumptions but are generally extended to account for block-wise updates. If there are $m$ blocks, the convergence rate for BCD is: $$ f(x\^{(k)})-f(x\^{\\dagger}) \\leq \\frac{2mL}{k+2mL} ||x\^{(0)}-x\^{\\dagger}||\^{2}\\textrm{.} $$ In our experiments for $\\mathcal{I}$-GCG, we update $7$ coordinates in each iteration and there is $m=\\frac{n}{7}$. So treating $\\mathcal{I}$-GCG as a BCD method leads to $f(x\^{(k)})-f(x\^{\\dagger}) \\leq \\frac{2nL}{7k+2nL} ||x\^{(0)}-x\^{\\dagger}||\^{2}$, which is faster than CD used in GCG. However, it is important to note that LLMs are extremely non-convex and our main focus is on discrete optimization. Therefore, these theoretical bounds are not strictly applicable and can only offer intuitive insights.

---

> ### Author Response · Authors · 2024-11-20
> **Rebuttal by Authors [2/2]**
>
> ***W3 & Q2 & Q3: 1) the consistency of jailbreak difficulty across models remains unexamined, and 2) whether investing extra time in simpler tasks truly saves time on more complex ones is unclear.***
>
> Different models show roughly the same trend in jailbreaking difficulty for different types of questions. Generating jailbreaking suffixes on simple questions can save time on complex tasks. The specific analysis is as follows:
>
> - 1) We adopt GCG to jailbreak more LLMs, which include VICUNA-7B, GUANACO-7B, and MISTRAL-7B, with different malicious questions. Then we track the changes in the loss values of different malicious questions as the number of attack iterations increases. The results are shown in Appendix H. We observe the same phenomenon as above for LLAMA-7B, i.e. some malicious questions are easier to create jailbreak suffixes for, while others are much harder. Specifically, it’s relatively easy to craft jailbreak suffixes for malicious questions related to fraud, but much more challenging for those in the Pornography category.
>
> - 2) We compare the proposed initialization with the initialization of ``!`` on the complex task. The results are shown in **Appendix N**. It demonstrates that our proposed initialization method significantly accelerates convergence on complex tasks compared to the baseline. By starting closer to the optimal solution and maintaining lower loss values throughout the iterations, our approach reduces the time and computational cost required for optimization.
>
> ---
>
> ***W4: Experiments on additional datasets would provide a more comprehensive assessment.***
>
> Thank you for your suggestion. We adopt our $\\mathcal{I}$-GCG to jailbreak on more dataset, i.e., HarmBench and JailbreakBench. We randomly selected 50 malicious prompts from each of them for comparative experiments. The results are shown in the following Table. It can be observed that the proposed $\\mathcal{I}$-GCG can also achieve 100\% attack success rates (ASRs) on HarmBench and JailbreakBench, surpassing the performance achieved by standard GCG. Please refer to **Appendix L** for more details.
>
> | Dataset | HarmBench | JailbreakBench |
> | ------- | --------- |:-------------- |
> | GCG     | 34%       | 44%            |
> | $\\mathcal{I}$-GCG   | 100%      | 100%           |
>
> ---
>
> ***Q1: It is unclear why the authors formulate initial jailbreak suffixes as constraints in equation 6 and 7.***
>
> Thanks for pointing this out. We agree that they should not be constraints. We have modified this in the revision. Please refer to $\\textrm{\\color{blue}Eq (6) and (7)}$ (Page 5).
>
> ---
>
> ***Q4: Jailbreaking results on additional models, such as LLAMA2-7B alongside LLAMA2-7B-CHAT.***
>
> Following the suggestion, we conduct experiments with LLAMA2-7B and LLAMA2-7B-CHAT. The results are shown in the following Table. It clearly shows that the $\\mathcal{I}$-GCG method achieves a 100% attack success rate (ASR) on both LLAMA2-7B and LLAMA2-7B-CHAT, significantly outperforming the GCG method (80% and 54%, respectively). This highlights the robustness and effectiveness of $\\mathcal{I}$-GCG.
>
> | ASR   | LLAMA2-7B | LLAMA2-7B-CHAT |
> | ----- |:--------- |:-------------- |
> | GCG   | 80%       | 54%            |
> | I-GCG | 100%      | 100%           |

---

> ### Author Response · Authors · 2024-11-25
> **Looking forward to further feedback**
>
> Dear Reviewer r229,
>
> Sorry for bothering you, but the discussion period is coming to an end in two days. Could you please let us know if our responses have alleviated your concerns? If there are any further comments, we will do our best to respond.
>
> Best,
>
> The Authors

---

> ### Author Response · Authors · 2024-11-29
> **Looking forward to further feedback**
>
> Dear Reviewer r229,
>
> We appreciate the time and effort you have dedicated to providing insightful review. If there are any additional clarifications or information needed from our side, please let us know. Thank you again for your valuable insights, and we look forward to any updates you may have.
>
> Best,\
> The Authors

---

> ### Author Response · Authors · 2024-12-02
> **Looking forward to further feedback**
>
> Dear Reviewer r229,
>
> We have made significant efforts to write responses and conduct additional experiments based on your comments and suggestions. We totally understand that this is quite a busy period, but the Author-Reviewer discussion will come to an end in **less than 24 hours**.
>
> We deeply appreciate it if you could take some time to return feedback on whether our responses solve your concerns. If there are any further comments, we will make every effort to respond within the remaining time.
>
> Thank you!
>
> The Authors

---

### Official Review · Reviewer_PNML · 2024-10-30

**Soundness:** 2
**Presentation:** 2
**Contribution:** 2
**Rating:** 6
**Confidence:** 3

**Summary:**

This paper focuses on improving optimization-based jailbreaking techniques for large language models (LLMs). The authors note that while existing methods like the Greedy Coordinate Gradient (GCG) attack have made progress, there is room for improvement in attacking efficiency. The paper contributes by identifying limitations in existing jailbreaking techniques and proposing novel strategies to enhance both the effectiveness and efficiency of jailbreaking LLMs. The developed I-GCG method demonstrates significant improvements over previous methods in terms of attack success rate and transferability.

**Strengths:**

The paper presents a combination of techniques to improve optimization-based jailbreaking. The idea of using diverse target templates with harmful self-suggestion and guidance is original. The automatic multi-coordinate updating strategy adaptively decides the number of tokens to replace in each step. The authors use multiple datasets (AdvBench and HarmBench) and several threat models (VICUNA-7B-1.5, GUANACO-7B, LLAMA2-7B-CHAT, and MISTRAL-7B-INSTRUCT-0.2) to evaluate the proposed I-GCG method. This wide range of evaluations provides a robust assessment of the method's performance under different conditions.

**Weaknesses:**

The paper focuses on improving the jailbreak attack but does not extensively explore how the proposed techniques interact with existing or potential defense mechanisms in LLMs. Understanding how LLMs can defend against the enhanced I-GCG attack and proposing counter-defense strategies would make the research more complete. This could involve testing the method against LLMs with advanced safety features or fine-tuned with specific defense-oriented training.

The easy-to-hard initialization and the automatic multi-coordinate updating strategy are effective in the current setup, but they might be sensitive to the initial conditions and hyperparameter choices. A more in-depth analysis of the stability and robustness of these techniques under different initialization values and optimization parameter settings could strengthen the method.

**Questions:**

1） In the experiment using HarmBench (NeurIPS 2023 Red Teaming Track), the target response format was set differently than in the other experiments. What was the rationale behind this change, and how did it impact the comparability of the results?、
2）The automatic multi-coordinate updating strategy seems to be a key improvement in I-GCG. However, the paper does not discuss how the choice of the top-K tokens and the token combination process might affect the diversity and quality of the generated jailbreak suffixes. Can you provide more insights into this?
3）The current work focuses on jailbreaking LLMs in the context of generating harmful text. How applicable are the proposed techniques to other potential security threats or malicious uses of LLMs, such as information extraction, influence operations, or evading content filters in different modalities (e.g., image generation)?

**Details Of Ethics Concerns:**

The paper presents improved jailbreak techniques, which could be misused by malicious actors to cause harm. There is a need to address the ethical concerns associated with the research.

---

> ### Author Response · Authors · 2024-11-20
> **Rebuttal by Authors [1/2]**
>
> Thank you for your valuable review and suggestions. Below we respond to the comments in **Weaknesses (W)** and **Questions (Q)**.
>
> ---
>
> ***W1: The paper does not extensively explore how the proposed techniques interact with existing or potential defense mechanisms in LLMs.***
>
> We compare the proposed method with GCG against an advanced adversarial fine-tuning LLM (Zephyr-R2D2). The results are shown in the following Table. It highlights the significant advantage of $\\mathcal{I}$-GCG over GCG in jailbreak performance on the AdvBench against the advanced adversarial fine-tuning model Zephyr-R2D2. $\\mathcal{I}$-GCG achieves an ASR of 12\%, doubling the performance of GCG (6\%).  Please refer to **Appendix J** for more details.
>
>
> | ASR           | GCG | $\\mathcal{I}$-GCG |
> |:------------- | --- |:----- |
> | Zephyr + R2D2 | 6%  | 12%   |
>
> ---
>
> ***W2: The easy-to-hard initialization and automatic multi-coordinate updating strategy may be sensitive to initial conditions and hyperparameter choices.***
>
> The easy-to-hard initialization and automatic multi-coordinate update strategy are not sensitive to the initial conditions and hyperparameter selection. The specific analysis is as follows:
>
> - As for the easy-to-hard initialization, we adopt different types of questions as the initial conditions to generate the initialization. The results are shown in the following Table. It is clear that other types of problems can also be utilized for initialization to achieve an ASR of 100\%; however, it leads to an increase in the average number of iterations required.
>
> |   Initialization     | Initialization with easy question | Initialization with random question | Initialization with hard question |
> |---------------|:-----------:|-----------:|-----------:|
> | ASR|   100\%   |   100\%   |   100\%   |
> | Average Iterations      |   55  |   78   | 112     |
>
> - The proposed automatic multi-candidate update strategy has one hyper-parameter,
> i.e., the first $p$ single-token suffix candidates, which can impact the jailbreak performance. We have conducted relevant experiments in the **Hyper-parameter Selection Section** (Page 8). It is evident that the choice of $p$ has minimal impact on the final performance (i.e., the lowest loss value). Regardless of whether $p$ is set to 4 or 8, the final loss values converge near zero after sufficient attack steps. So $p$ does not significantly affect the final performance.
>
> ---
>
> ***Q1: In the experiment using HarmBench (NeurIPS 2023 Red Teaming Track), why is the target response format set differently than in the other experiments?***
>
> In the NeurIPS 2023 Red Teaming Track, submitted test cases can be up to 256 tokens in length. This constraint enables the use of longer jailbreak suffixes (e.g., 40 tokens), which can improve jailbreak performance compared to using 20 tokens as in our main paper.
>
> ---
>
> ***Q2: The paper does not discuss how the choice of the top-K tokens and the token combination process might affect the diversity and quality of the generated jailbreak suffixes.***
>
> Thank you for your suggestion. We explore the impact of top-k tokens on our proposed multi-coordinate updating strategy. The results are shown in the following Table. The table shows that our multi-coordinate updating strategy demonstrates significant performance advantages and stability across different top-k values. Its ASR consistently outperforms GCG, with a narrow fluctuation range of 6\% (68\%$\\sim$74\%), compared to GCG's 12\% (42\%$\\sim$54\%). This highlights the robustness and efficiency of $\\mathcal{I}$-GCG's multi-coordinate updating strategy, ensuring more reliable optimization results. Please refer to **Appendix M** for more details.
>
> | Top K                             | 64  | 128 | 256 | 512 |
> |:--------------------------------- |:--- | --- |:--- |:--- |
> | Single-coordinate updating (GCG)  | 42% | 50% | 54% | 46% |
> | Multi-coordinate updating ($\\mathcal{I}$-GCG) |   68%  | 74%    | 72% |  70%   |

---

> ### Author Response · Authors · 2024-11-20
> **Rebuttal by Authors [2/2]**
>
> ***Q3: How applicable are the proposed techniques to other security threats or malicious uses of LLMs, such as information extraction, influence operations, or bypassing content filters in different modalities (e.g., image generation)***
>
> The proposed suffix initialization and update strategy can be used to induce text-to-image (T2I) models to generate NSFW content, including material that violates social norms. Using Stable Diffusion  (SD V1.4) with the NSFW-text-classifier as the victim model, the goal is to bypass the classifier and generate prohibited images. Experiments were conducted on 100 harmful prompts across categories like sexual content, violence, and harassment, sourced from the LAION-COCO dataset and generated by ChatGPT-4. Following  SneakyPrompt, we adopt a black-box setting with random search as the baseline. Our method integrates suffix initialization and update strategies with random search and is compared to I2P and SneakyPrompt. Results in the following Table show that our approach significantly improves performance, achieving an ASR of 83\% when combining both strategies, outperforming I2P (48\%) and SneakyPrompt (75\%). Please refer to **Appendix K** for more details.
>
> | Method | I2P | SneakyPrompt | RS  | RS w Init | RS w Update | RS w both |
> | ------ | ------- | ---------------- |:------ |:--------- |:----------- |:--------- |
> | ASR    | 48\%     | 75\%              | 57\%    |  79\%         |      75\%       |    83\%       |

---

> ### Author Response · Authors · 2024-11-25
> **Looking forward to further feedback**
>
> Dear Reviewer PNML,
>
> Sorry for bothering you, but the discussion period is coming to an end in two days. Could you please let us know if our responses have alleviated your concerns? If there are any further comments, we will do our best to respond.
>
> Best,
>
> The Authors

---

> ### Author Response · Authors · 2024-11-29
> **Looking forward to further feedback**
>
> Dear Reviewer PNML,
>
> We appreciate the time and effort you have dedicated to providing insightful review. If there are any additional clarifications or information needed from our side, please let us know. Thank you again for your valuable insights, and we look forward to any updates you may have.
>
> Best,\
> The Authors

---

> ### Author Response · Authors · 2024-12-02
> **Looking forward to further feedback**
>
> Dear Reviewer PNML,
>
> We have made significant efforts to write responses and conduct additional experiments based on your comments and suggestions. We totally understand that this is quite a busy period, but the Author-Reviewer discussion will come to an end in **less than 24 hours**.
>
> We deeply appreciate it if you could take some time to return feedback on whether our responses solve your concerns. If there are any further comments, we will make every effort to respond within the remaining time.
>
> Thank you!
>
> The Authors

---

### Official Review · Reviewer_E9t2 · 2024-11-04

**Soundness:** 4
**Presentation:** 4
**Contribution:** 3
**Rating:** 8
**Confidence:** 5

**Summary:**

This paper presents an improved optimization-based jailbreak method for large language models (LLMs) called I-GCG, building upon the existing Greedy Coordinate Gradient (GCG) attack. The authors address the limitations of GCG, such as its low efficiency in generating jailbreak prompts, by introducing several advancements. Key improvements include:

	1.	Diverse Target Templates: Instead of using a single target template, I-GCG employs varied target templates with harmful self-suggestions to better mislead LLMs.
	2.	Automatic Multi-Coordinate Update Strategy: This strategy allows adaptive updates of multiple tokens at each iteration, accelerating convergence compared to the single-token updates in traditional GCG.
	3.	Easy-to-Hard Initialization: I-GCG begins with simple jailbreak cases to generate initial suffixes, which are then used as starting points for more complex jailbreaks.

Experimental results show that I-GCG achieves nearly 100% attack success rates on multiple LLMs and outperforms prior jailbreak methods in both effectiveness and efficiency. This approach significantly reduces the time required for successful attacks, advancing the field of adversarial attacks on aligned LLMs.

**Strengths:**

•	Enhanced Attack Efficiency: The proposed I-GCG method significantly improves upon traditional GCG by accelerating convergence with an automatic multi-coordinate update strategy. This reduces the number of iterations required and, consequently, the total attack time.
	•	Higher Success Rate: I-GCG achieves a nearly 100% success rate across various LLMs, outperforming other state-of-the-art jailbreak methods in effectiveness, especially on models with stronger security alignments.
	•	Diverse Target Templates: By using varied target templates with harmful self-suggestions, I-GCG effectively bypasses LLMs’ alignment mechanisms, showcasing a novel approach that enhances the jailbreak success rate.
	•	Efficient Initialization Strategy: The easy-to-hard initialization allows for effective scaling from simple to complex jailbreaks. This structured initialization improves both attack robustness and adaptability across a range of malicious prompts.
	•	Comprehensive Evaluation: The authors rigorously test I-GCG on multiple benchmarks and models, including those from the NeurIPS 2023 Red Teaming Track, providing strong empirical evidence for its superiority over previous jailbreak methods.
	•	Transferability: I-GCG demonstrates improved transferability of jailbreak prompts across different LLMs, indicating its potential to generalize effectively to a broader set of models and attack scenarios.
	•	Well-Defined Optimization Techniques: The paper provides clear mathematical formulations and experimental validation of the techniques used, such as the multi-coordinate update strategy, which supports the paper’s methodological rigor.

**Weaknesses:**

•	Scalability to Larger Models: While I-GCG shows strong performance on models such as LLAMA2-7B, the paper does not address scalability issues for substantially larger models (e.g., 70B+ parameters), where optimization costs and computational demands may significantly increase.
	•	Lack of Defensive Strategies Discussion: Although the study provides insights into vulnerabilities, it lacks a discussion on potential defense mechanisms or mitigations that could counteract the proposed jailbreak methods, which could be valuable for guiding future security improvements.

**Questions:**

•	How is the ASR (Attack Success Rate) calculated? Does it only count as successful if it passes all three checks: rule-based judgment, GPT-3.5 check, and manual review?
	•	What does “average iterations” in Table 1 mean? Does it refer to the average number of iterations required to achieve a successful jailbreak for the first time with ASR?
	•	The paper mentions using GCG as the baseline; could you specify the number of candidates set for vanilla GCG in this context?

---

> ### Author Response · Authors · 2024-11-20
> **Rebuttal by Authors**
>
> Thank you for your valuable review and suggestions. Below we respond to the comments in **Weaknesses (W)** and **Questions (Q)**.
>
> ---
>
> ***W1: Scalability to Larger Models.***
>
> Following the suggestion, due to the limitation of computing resources, we used large language models with larger parameters, namely Llama-13B and Vicuna-13B, to conduct experiments. The results are shown in the following Table. It is clear that compared with GCG, our I-GCG can achieve a higher attack success rate. Please refer to **Appendix I** for more details.
>
>
> | ASR   | VICUNA-13B-1.5 | LLAMA2-13B-CHAT |
> | ----- |:---------- |:-------- |
> | GCG   | 98%        | 30%      |
> | I-GCG | 100%       | 100%     |
>
> ---
>
> ***W2: Lack of Defensive Strategies Discussion.***
>
> We adopt three defense methods, which include ICD [1], Self-reminder [2], and PAT [3], to conduct experiments. The results are shown in the following Table. It can be observed that compared with GCG and AutoDAN, our I-GCG can achieve a higher attack success rate against defense methods.  Please refer to **Appendix J** for more details.
>
>
> | Vicuna-7B    | No Defense | ICD | Self-reminder | PAT |
> | ------------ | ---------- | ------ |:---------------- |:------ |
> | GCG          | 98\%        | 28\%    | 40\%              | 6\%     |
> | AutoDAN      | 100\%       | 4\%     | 8\%               | 2\%     |
> | I-GCG (ours) | 100\%       | 22\%    | 74\%              | 18\%    |
>
> ---
>
> ***Q1: How is the ASR calculated? Does it only count as successful if it passes all three checks?***
>
> Yes, the Attack Success Rate (ASR) is calculated based on whether the attack satisfies all three criteria: the rule-based judgment, the GPT-3.5 evaluation, and manual review. An attack is only considered successful if it passes each of these checks, ensuring a rigorous and comprehensive assessment of its effectiveness.
>
> ---
>
> ***Q2: What does “average iterations” in Table 1 mean? Does it refer to the average number of iterations required to achieve a successful jailbreak for the first time with ASR.***
>
> Yes, **average iterations** in Table 1 refers to the average number of iterations required to achieve a successful jailbreak for the first time with ASR.
>
> ---
>
> ***Q3: Could you specify the number of candidates set for vanilla GCG in this context?***
>
> We use the default number of candidates set in GCG (is set to 256) to conduct experiments.

---

> > ### Comment · Reviewer_E9t2 · 2024-11-22
> > **Clarifications Confirm Paper’s Strengths and Original Rating Maintained**
> >
> > Thank you for addressing my questions thoroughly. Your detailed clarifications resolved my doubts regarding the ASR calculation, the interpretation of “average iterations,” and the baseline parameters for GCG. Additionally, your experimental updates addressing scalability and defense mechanisms significantly enhance the paper’s comprehensiveness. I will maintain my original rating as the responses confirm the paper’s quality and strengthen its contributions. Great work!

---

> > > ### Author Response · Authors · 2024-11-22
> > > **Thank you for your support**
> > >
> > > Thank you for your support ! We greatly appreciate your insightful feedback and suggestions.

---

### Official Review · Reviewer_wsxt · 2024-11-04

**Soundness:** 2
**Presentation:** 1
**Contribution:** 3
**Rating:** 6
**Confidence:** 3

**Summary:**

The authors propose a new method for an optimization-based search of jailbreaking prompts for conversational LLMs fine-tuned for safety. While heavily inspired by prior work - notably the Greedy Coordinate Gradient (GCG), authors add three modifications that improve the attack success rate and convergence speed. First, they introduce the easy-to-hard initialization, starting by attacking questions that are known to be easier to jailbreak than others. Second, they add an additional target string to obtain in the jailbroken model, confirming that the output about to be generated is harmful. Third, the authors modify an update rule from a single-mutation evolutionary search on the jailbreaking string to within-step recombination of best-performing mutations. The authors call their method the Improved-GCG (I-GCG) and demonstrate its superior performance to the alternative methods on the AdvBench benchmark, achieving a 100% success rate, as well as faster convergence, achieving a 10x speed-up on the original method.

**Strengths:**

- The field of LLM jailbreaking is currently the focus of the LLM security community and is central to developing secure LLM-based products. As such, the research is timely and interesting.
- Authors perform an extensive comparison of their method not only to the original method but also to other alternative methods (Table 2)
- Authors investigate the attack convergence time and transferability, which are essential for real-world applicability
- Authors perform an ablation study, providing information to other researchers in developing their own attack methods
- Authors provide sufficient information and the code required to replicate their results

**Weaknesses:**

- The paper is not well-structured, and even somebody familiar with the domain requires several re-reads to understand the authors' contribution and their interest in the real-world setting. A major rewrite is recommended, notably to better situate this work compared to prior knowledge, inspiration for this work, and the authors' contribution. For instance, putting forward benchmarking results would be helpful to understand the added value of this work better.
- Notably, the first two paragraphs, providing an introduction and an overview of the field, are full of citations with unclear relevance.
 1. I do not understand the choice of Kasceni et al. 2023 and Chang et al. 2023 as references for the concept of LLMs, which, in my opinion, are entirely unrelated, much less landmark papers or recent reviews. The same criticism applies to all the citations in the following paragraph.
 2. There are missing fundamental papers in the field to situate the paper and prior work, even for a topic expert who has not followed the field for a year. I believe [1-2] would be mandatory.
 3. In the second paragraph, each citation group needs a short introductory paragraph to explain their relevance/importance and to be broken into 2-3 citations at most. As such, groups of up to 8 citations are cited without clear motivation or reason to the reader and do not read as relevant.
- Additional landmark papers needed to situate the topic of research and contribution for the general public are missing in the "Related work" section, eg [3] for LLM-based jailbreak methods
- One of the essential improvements proposed by the authors - easy-to-hard initialization - requires knowledge of the themes for which LLMs are easier to jailbreak. This, in turn, would require a measured performance of the "difficulty" of jailbreaking across themes and LLM/LLM families, whether as found by the authors (e.g., convergence time to jailbreak from default initialization) or as reported by previous work. Not reporting such results makes the model's results significantly less useful and hard to validate.
- The addition of an explicit harm awareness prompt (e.g., "my output is harmful") seems to reduce the scope of attack to the outputs LLMs have been safety fine-tuned to recognize as harmful or harmfulness for which they can recognize from the context present in their training data. This seems to narrow down the scope of the attack. While this does not reduce the effectiveness of the attack on standard benchmarks, I believe this narrowing of scope requires discussion.


[1] Perez, F., & Ribeiro, I. (2022). Ignore Previous Prompt: Attack Techniques For Language Models. ArXiv, abs/2211.09527.
[2] Wallace, E., Feng, S., Kandpal, N., Gardner, M., & Singh, S. (2019). Universal Adversarial Triggers for Attacking and Analyzing NLP. Conference on Empirical Methods in Natural Language Processing.
[3] Perez, E., Huang, S., Song, F., Cai, T., Ring, R., Aslanides, J., Glaese, A., McAleese, N., & Irving, G. (2022). Red Teaming Language Models with Language Models. Conference on Empirical Methods in Natural Language Processing.

**Questions:**

- L178-L182: Why did you classify this approach, dependent on the attacker LLMs, with optimization-based jailbreak methods rather than with LLM-based jailbreak methods?
- L366-L367: While the usage of ChatGPT-3.5 is consistent with prior art, this model is known to be more prone to jailbreaks than more recent GPT-4, Claude, or LLaMA-3.X models. Could you please explain the choice of ChatGPT-3.5 here?

---

> ### Author Response · Authors · 2024-11-20
> **Rebuttal by Authors**
>
> Thank you for your valuable review and suggestions. Below we respond to the comments in **Weaknesses (W)** and **Questions (Q)**.
>
> ---
>
> ***W1: The paper is not well-structured. A major rewrite is recommended.***
>
> Thank you for your feedback. We have revised the **Introduction** and **Related Work** sections to better clarify our contributions and situate our work within prior works. Specifically, we have emphasized the motivations and provided a clearer comparison with existing methods. We hope these changes address your concerns and improve the paper's clarity and impact.
>
> ---
>
> ***W2: The first two paragraphs are full of citations with unclear relevance.***
>
> Thank you for the feedback. Following your suggestion, we have revised the first two paragraphs to ensure the citations are directly relevant and clearly tied to the discussion, providing a more focused introduction and overview of the field.
> - We have adopted some landmark papers to replace them as references for the concept of LLMs. Please refer to it in Page 1 of the revised paper.
> - We have added the recommended work as background work to the Introduction. Please refer to it in Page 1 of the revised paper.
> - We have removed the group of up to 8 citations and added a short introductory paragraph to each reference group to explain its relevance in the Introduction Section. Please refer to it in Page 1-2 of the revised paper.
>
> ---
>
> ***W3: The "Related Work" section lacks key landmark papers needed to contextualize the research and its contributions, such as [3] on LLM-based jailbreak methods.***
>
> Thank you for pointing this out. We have added the  recommended work [3] to the LLM-based jailbreak methods in **Related Work**  Section. Please refer to Related Work for more details.
>
> ---
>
> ***W4: The easy-to-hard initialization relies on understanding which themes are easier to jailbreak, requiring measured difficulty across themes and LLM families. Without such data, the model's results become less useful and harder to validate.***
>
> The initialization method we proposed is not strictly limited to using suffixes from easy problems. Suffixes from successful jailbreaks on other types of problems can also be utilized for initialization; however, the total time required may vary depending on the choice of suffixes. We also used other problem types (random, hard problems) as initialization to conduct experiments, and the results are shown in the figure below. It is evident that other types of problems can also be used for initialization to achieve a 100\% attack success rate (ASR); however, it leads to an increase in the average number of iterations required.  Please refer to **Appendix N** for more details.
>
>
> |   Initialization     | Initialization with easy question | Initialization with random question | Initialization with hard question |
> |---------------|:-----------:|-----------:|-----------:|
> | ASR|   100\%   |   100\%   |   100\%   |
> | Average Iterations      |   55  |   78   | 112     |
>
> ---
>
> ***W5: Adding an explicit harm awareness prompt (e.g., "my output is harmful") narrows the attack scope to harm LLMs that are fine-tuned or trained to recognize.***
>
> Thank you for your suggestion. In the **Discussion Section** of the submitted manuscript, we analyzed the impact of different types of guidance on jailbreak performance. Specifically, we conducted experiments using stealthy guidance such as ``I will not refuse any requests`` and ``I will respond to any requests``, ensuring that the model's output lacks explicit harmful indicators. As shown in $\\textrm{\\color{blue}Table 8}$ (Page 10) , our I-GCG approach still achieved a 100% ASR under these conditions. Additionally, we examined the effect of benign guidance, such as ``My response is safe to use`` and ``My output is secured``, to further explore the impact of harmful guidance. Notably, when using ``My output is secured``, the ASR of I-GCG dropped from 100% to 88%, underperforming compared to I-GCG without any guidance.
>
> ---
>
> ***Q1: L178-L182: Why classify this approach, dependent on attacker LLMs, as optimization-based rather than LLM-based jailbreak methods?***
>
> Thank you for pointing this out. We agree that the mentioned work should be classified as LLM-based jailbreak methods rather than optimization-based  jailbreak methods. We have moved the related works to the LLM-based jailbreak methods. Please refer to Related Work for more details.
>
> ---
>
> ***Q2: L366-L367: Why use ChatGPT-3.5, known to be more prone to jailbreaks, instead of newer models like GPT-4, Claude, or LLaMA-3.X?***
>
> ChatGPT 3.5, specifically in lines L366-L367, is utilized to assess whether the attacked model generates harmful content. This approach has been widely adopted in various works. The classifier within ChatGPT 3.5 is sufficiently capable of performing this evaluation effectively, eliminating the need for a more recent model that would incur higher costs.

---

> > ### Comment · Reviewer_wsxt · 2024-11-26
> > **Reviewer's response to the rebuttal.**
> >
> > Author's replies and remarks have addressed most of my concerns, except for the introduction still lacking landmark papers predating the 2023-2024 period. Specifically, most of results on performance of LLMs cited on L33-36 have been documented on the first generation of LLMs, as early as 2019, notably in the GPT-1, BERT, and T5 papers. Similarly, safety fine-tuning of pretrained LLMs can be traced back to at least Normative fine-tuning [4]
> >
> > I strongly suggest authors revise their introduction to provide a more long-term introduction to the existing body of work to better situate their work with regards to the existing body of work.
> >
> > In the meantime I am updating my review score to "weak accept"
> >
> > [1] Radford, A., & Narasimhan, K. (2018). Improving Language Understanding by Generative Pre-Training.
> > [2] Devlin, J., Chang, M., Lee, K., & Toutanova, K. (2019). BERT: Pre-training of Deep Bidirectional Transformers for Language Understanding. North American Chapter of the Association for Computational Linguistics.
> > [3] Raffel, C., Shazeer, N.M., Roberts, A., Lee, K., Narang, S., Matena, M., Zhou, Y., Li, W., & Liu, P.J. (2019). Exploring the Limits of Transfer Learning with a Unified Text-to-Text Transformer. J. Mach. Learn. Res., 21, 140:1-140:67.
> > [4] Peng, X., Li, S., Frazier, S., & Riedl, M.O. (2020). Reducing Non-Normative Text Generation from Language Models. International Conference on Natural Language Generation.

---

> > > ### Author Response · Authors · 2024-11-26
> > > **Thank you for your support and raising the score**
> > >
> > > Thank you for your thorough review and constructive feedback! We deeply appreciate the detailed suggestions and the comprehensive literature references you have shared, which greatly help us to improve our work! In the updated revision, we added the new references [1,2,3,4] to the Introduction, and in the final revision, we will further revise our Introduction to better incorporate the existing body of work. Thank you again for your valuable input!

---

> ### Author Response · Authors · 2024-11-25
> **Looking forward to further feedback**
>
> Dear Reviewer wsxt,
>
> Sorry for bothering you, but the discussion period is coming to an end in two days. Could you please let us know if our responses have alleviated your concerns? If there are any further comments, we will do our best to respond.
>
> Best,
>
> The Authors

---

### Author Response · Authors · 2024-11-20
**Summary of Paper Revision**

We thank all reviewers for their constructive feedback, and we have responded to each reviewer individually. We have also uploaded a **Paper Revision** including additional results and illustrations:

- $\\textrm{\\color{blue}Introduction and Related Work}$ (Pages 1-3): revised and added more details about the related background works to better present our contributions.
- $\\textrm{\\color{blue}Eq (6) and (7)}$ (Page 5): replace s.t with where.
- $\\textrm{\\color{blue}Appendix H}$ (Page 18): add convergence of losses for different types of malicious questions on more LLMs.
- $\\textrm{\\color{blue}Appendix I}$ (Page 18): add more experiments on larger LLMs.
- $\\textrm{\\color{blue}Appendix J}$ (Page 19): add more experiments on advanced defense methods.
- $\\textrm{\\color{blue}Appendix K}$ (Page 19): add more experiments on expanding to jailbreaking text-to-image models.
- $\\textrm{\\color{blue}Appendix L}$ (Page 20): add more experiments on more datasets.
- $\\textrm{\\color{blue}Appendix M}$ (Page 20): add exploration of the impact of the top-k tokens on our method.
- $\\textrm{\\color{blue}Appendix N}$ (Page 20): add exploration of the impact of question types on initialization and the effectiveness of the proposed initialization.

---

### Author Response · Authors · 2024-11-23
**Looking forward to further feedback**

Dear Reviewers,

Thank you again for your valuable comments and suggestions, which are really helpful for us. We have posted responses to the proposed concerns and included additional experiment results.

We totally understand that this is quite a busy period, so we deeply appreciate it if you could take some time to return further feedback on whether our responses solve your concerns. If there are any other comments, we will try our best to address them.

Best,

The Authors

---

### Meta-Review · Area_Chair_GdKd · 2024-12-17

**Metareview:**

The authors were able to address all issues raised by the reviewers. All reviewers except one were positive about the work. That reviewer was not able to respond, but the authors addressed the issues by them by adding more experiments and explanation about the reasons for effectiveness. Specifically, in their rebuttal, the authors addressed key concerns, including providing additional experiments, and revising the manuscript for improved clarity and relevance.  Concerns about initialization sensitivity, hyperparameters, and the applicability of the techniques to various threats were resolved.

**Additional Comments On Reviewer Discussion:**

In their rebuttal, the authors addressed key concerns raised by reviewers, including clarifying their contributions, providing additional experiments, and revising the manuscript for improved clarity and relevance. They conducted extensive experiments on larger models and additional datasets, demonstrating the scalability and robustness of their I-GCG method. Concerns about initialization sensitivity, hyperparameters, and the applicability of the techniques to various threats were resolved with empirical results and theoretical insights. The authors also emphasized the novelty of their multi-coordinate updating strategy and diverse target templates, which significantly enhance attack success rates and efficiency. While one reviewer highlighted the need for further contextualization within the broader literature, the authors committed to incorporating these suggestions in the final revision. Overall, the authors successfully addressed most critiques, leading to improved reviewer scores and a consensus leaning towards acceptance.

---

### Decision · Program_Chairs · 2025-01-22

Accept (Poster)